# Arsenic Trioxide and Venetoclax Synergize against AML Progenitors by ROS Induction and Inhibition of Nrf2 Activation

**DOI:** 10.3390/ijms23126568

**Published:** 2022-06-12

**Authors:** Dinh Hoa Hoang, Ralf Buettner, Melissa Valerio, Lucy Ghoda, Bin Zhang, Ya-Huei Kuo, Steven T. Rosen, John Burnett, Guido Marcucci, Vinod Pullarkat, Le Xuan Truong Nguyen

**Affiliations:** 1Gehr Family Center for Leukemia Research, Hematology Malignancies and Stem Cell Transplantation Institute, City of Hope National Medical Center, Duarte, CA 91010, USA; hhoang@coh.org (D.H.H.); rbuettner@coh.org (R.B.); mvalerio@coh.org (M.V.); lghoda@coh.org (L.G.); bzhang@coh.org (B.Z.); ykuo@coh.org (Y.-H.K.); srosen@coh.org (S.T.R.); gmarcucci@coh.org (G.M.); 2Center for Gene Therapy, Beckman Research Institute, City of Hope National Medical Center, Duarte, CA 91010, USA; jburnett@coh.org

**Keywords:** acute myeloid leukemia (AML), arsenic trioxide (ATO), venetoclax (VEN), oxidative phosphorylation (OXPHOS), nuclear factor erythroid 2-related factor 2 (Nrf2), reactive oxygen species (ROS)

## Abstract

Venetoclax (VEN) in combination with hypomethylating agents induces disease remission in patients with de novo AML, however, most patients eventually relapse. AML relapse is attributed to the persistence of drug-resistant leukemia stem cells (LSCs). LSCs need to maintain low intracellular levels of reactive oxygen species (ROS). Arsenic trioxide (ATO) induces apoptosis via upregulation of ROS-induced stress to DNA-repair mechanisms. Elevated ROS levels can trigger the Nrf2 antioxidant pathway to counteract the effects of high ROS levels. We hypothesized that ATO and VEN synergize in targeting LSCs through ROS induction by ATO and the known inhibitory effect of VEN on the Nrf2 antioxidant pathway. Using cell fractionation, immunoprecipitation, RNA-knockdown, and fluorescence assays we found that ATO activated nuclear translocation of Nrf2 and increased transcription of antioxidant enzymes, thereby attenuating the induction of ROS by ATO. VEN disrupted ATO-induced Nrf2 translocation and augmented ATO-induced ROS, thus enhancing apoptosis in LSCs. Using metabolic assays and electron microscopy, we found that the ATO+VEN combination decreased mitochondrial membrane potential, mitochondria size, fatty acid oxidation and oxidative phosphorylation, all of which enhanced apoptosis of LSCs derived from both VEN-sensitive and VEN-resistant AML primary cells. Our results indicate that ATO and VEN cooperate in inducing apoptosis of LSCs through potentiation of ROS induction, suggesting ATO+VEN is a promising regimen for treatment of VEN-sensitive and -resistant AML.

## 1. Introduction

The quiescent leukemia stem cell population (LSC) plays a pivotal role in disease persistence and drug resistance in AML [1]. LSCs have unique features, including the need to maintain low reactive oxygen species (ROS) and high Bcl-2 levels, which could be exploited for pharmacological targeting [2,3]. In addition, LSCs depend on oxidative phosphorylation (OXPHOS) for energy production, while normal hematopoietic stem cells (HSCs) mainly utilize glycolysis [4,5,6]. The Bcl-2 family of proteins mediates antioxidant functions and regulates mitochondria metabolism including ROS and OXPHOS [7]. Venetoclax (VEN) is a selective Bcl-2 inhibitor that induces ROS, blocks OXPHOS and causes apoptosis in LSCs [2]. While single agent VEN shows only modest clinical efficacy in AML, high response rates are obtained in AML patients treated with VEN in combination with hypomethylating agents (HMAs) or low dose cytarabine [2,8]. However, a significant proportion of AML patients do not respond or acquire resistance to current VEN-based therapies [9,10]. Of note, recent studies reported that LSCs from patients with refractory/relapsed AML are less sensitive to VEN-based treatment regimens since these cells can also utilize fatty acid oxidation (FAO) to feed into OXPHOS [11,12]. Therefore, novel VEN-based combinations are urgently needed to further potentiate VEN-based therapy and overcome resistance to Bcl-2 inhibition by targeting metabolic features of LSCs.

Arsenic trioxide (As_2_O_3_, ATO) induces the death of leukemic cells in part through generation of ROS and induction of DNA damage [13]. Although remarkably effective in acute promyelocytic leukemia (APL), ATO has minimal clinical activity in other AML subtypes [14,15,16,17,18]. Previous studies have suggested that non-APL AML cells, compared to APL cells, generally accumulate lower intracellular amounts of ATO and have demonstrated that low intracellular accumulation of ATO can be attributed to low expression of the arsenic transporter Aquaporin-9 (AQP9), thus making these cells less sensitive to ATO treatment [19]. A positive correlation of AQP9 expression and ATO-induced cytotoxicity has been described for 10 out of 11 myeloid and lymphod cell lines [20]. Upregulation of AQP9 by G-CSF or Azacytidine enhanced the effect of ATO in non-APL AML cells [19,21]. In addition, several other arsenic-based combination studies, including ATO-tyrosine kinase inhibitor combinations, have been conducted in vitro and in vivo with the aim of overcoming ATO-resistance in non-APL AML cells [13,22,23,24,25,26,27,28]. Increases in ROS levels in cells are normally tightly controlled by an inducible antioxidant program to counterbalance high ROS levels in order to maintain redox balance [29,30]. The major pathway by which cells counteract high ROS levels is through the activation of the nuclear factor erythroid 2-related factor 2 (Nrf2) antioxidant pathway [31,32,33]. Nuclear translocation of the transcription factor Nrf2 leads to the transcription of antioxidant gene products in cells. The loss of Nrf2 in cancer cells increases oxidative stress, which can result in diminished tumorigenesis [34]. All trans retinoic acid (ATRA), ATO and the combination of ATRA plus ATO are effective therapeutic regimens for the treatment of APL [16,18,35]. The augmented cytotoxicity of the ATO-ATRA combination in APL cells, compared to a single agent treatment, has been attributed, at least in part, to the inhibition of the Nfr2 antioxidant pathway [36]. The ATO plus ATRA combination also demonstrated preclinical activity against NPM1 mutant AML cells [37,38], and clinical trials with this combination are currently under way in non-APL AML patients, including in NPM1 mutant AML (clinical trials identifiers NCT03031249 and NCT05297123). We have previously shown that VEN is a potent inhibitor of Nrf2 activation [31]. ATO is generally well tolerated by patients and its side effects are manageable [39]. We hypothesized that ATO would synergize with VEN and examined the in vitro activity of the ATO+VEN combination against AML progenitor cells.

## 2. Results and Discussion

### 2.1. ATO Induces ROS and p-AKT in LSC-Enriched AML Progenitor Cells

The Nrf2 antioxidant pathway is implicated in mediating chemoresistance in AML and could particularly protect LSCs given their dependence on Nrf2 to maintain low ROS levels [40]. Therefore, activation of Nrf2 by ATO-induced ROS could attenuate the cytotoxicity of ATO. Normally, the transcription factor Nrf2 is kept inactive through binding to its inhibitor, Kelch-like ECH-associated protein 1 (Keap1), subsequently targeting Nrf2 for proteasomal degradation [32,41,42,43]. During oxidative stress, elevated amounts of ROS oxidize redox sensitive cysteine residues of the Keap1 protein, resulting in release of Nrf2 from Keap1, Nrf2 nuclear translocation and transcription of antioxidant enzymes, such as heme oxygenase-1 (HO-1) and NADP-quinone oxidoreductase-1 (NQO-1), which are involved in ROS neutralization [31,44]. Activation of the PI3K/Akt pathway has a profound effect on the cellular redox status, and activation of Akt can further augment Nrf2 pathway activation [45]. Given that the cytotoxicity of various anti-AML therapeutics partly depends on ROS levels, we examined the effects of ATO on ROS in LSC-enriched progenitor cells. Treatment of LSC-enriched progenitors with 500 nM ATO did not significantly increase mitochondrial ROS levels in the first 3 h, but strongly induced mitochondrial ROS after 12 h (Figure 1A). It has been reported that ATO inhibits Akt activation in several solid tumor and leukemic cells [15,46]. We, however, observed that treatment of the LSC-enriched progenitor cells with 500 nM ATO transiently increased p-Akt expression between 0.5 and 4 h and p-Akt levels return to baseline after 12 h (Figure 1B). Co-treatment with the Akt inhibitor LY294002 (10 µM) reversed the effect of ATO on early activation of Akt (Figure 1C).

### 2.2. ATO-Induced ROS Activation Is Attenuated by Concomitant Activation of the p-Akt/Nrf2 Antioxidant Pathway in LSC-Enriched AML Progenitor Cells

The effects of ATO on ROS induction and Akt phosphorylation seen in LSC-enriched AML cells prompted us to hypothesize that early activation of Akt by ATO may attenuate ROS induction by ATO, given the known effects of Akt phosphorylation on Nrf2 activation [45]. Using immunofluorescence and a cellular fractionation assay, we demonstrated that treatment of LSC-enriched progenitor cells with ATO induced Nrf2 translocation to the nucleus whereas knockdown (KD) or pharmacological inhibition of Akt with LY294002 reversed this effect (Figure 2A,B). These results demonstrate that early activation of Akt by ATO induces nuclear translocation of Nrf2, followed by Nrf2-induced transcription of antioxidant enzymes, which are known to attenuate ROS levels [31,44]. We next explored the mechanism of ATO-induced Nrf2 translocation. Previous studies showed that PI3K/Akt activation causes actin cytoskeleton rearrangement in response to oxidative stress, leading to Nrf2/actin complex translocation into the nucleus [47]. Of note, we observed that treatment with ATO induced actin translocation to the nucleus (Figure 2B), suggesting that ATO regulates nuclear translocation of the Nrf2/actin complex through Akt activation. Indeed, immunoprecipitation of nuclear extracts from LSC-enriched progenitor cells demonstrated that ATO enhanced the interaction of Nrf2 and actin in the nucleus, while co-treatment with AKT-siRNA or Akt inhibitor LY294002 disrupted this interaction (Figure 2C). As expected, ATO dissociated the Nrf2/Keap1 interaction (Figure 2D), resulting in Nrf2 nuclear translocation and decreased Nrf2 ubiquitination (Figure 2E). Knockdown (KD) of Akt with siRNA or pharmacological inhibition of Akt with LY294002 abolished these effects (Figure 2D,E). ATO-induced nuclear translocation and stabilization of Nrf2 resulted in increased transcription of the antioxidant enzymes NQO-1 and HO-1 (Figure 2F). Knockdown of Akt or Nrf2 by siRNA disrupted ATO-induced NQO-1 and HO-1 expression (Figure 2F), leading to increased levels of ROS (Figure 2G) and apoptosis, as demonstrated by increased Annexin V staining and induction of DNA fragmentation (Figure 2H). These results demonstrate that ATO-mediated apoptosis of LSC-enriched progenitor cells was attenuated by Nrf2 activation through ATO-induced ROS as well as early activation of Akt. Pharmacologic Akt inhibition could potentiate ATO cytotoxicity by inhibiting Nrf2 activation. Of note, synergistic killing of AML cells with ATO and Akt inhibitor LY294002 has previously been demonstrated. AML cells exhibited significantly reduced sensitivity to ATO when attached to human stromal cells; however, sensitivity to ATO was restored, at least in part, through inhibition of Akt [27].

### 2.3. Addition of VEN to ATO Inhibits Nrf2 Nuclear Translocation, Augments ROS and Increases Apoptosis in LSC-Enriched AML Progenitor Cells

We previously showed that VEN is a potent inhibitor of Nrf2 activation induced by HMA [31]. We hypothesized that VEN could also inhibit ATO-induced Nrf2 activation and potentiate ATO-mediated cytotoxicity and thus examined the effects of the ATO+VEN combination on LSC-enriched AML progenitor cells. Our results demonstrate that addition of VEN to ATO blocked ATO-induced Nrf2 translocation to the nucleus in LSC-enriched AML progenitor cells, as shown by immunofluorescence labeling and cytoplasm-nuclear fractionation assays (Figure 3A). Addition of VEN also reversed the ATO-induced Nrf2-Keap1 dissociation, leading to Keap1-induced Nrf2 ubiquitination and degradation (Figure 3B). The addition of VEN also abolished ATO-Nrf2-induced upregulation of NQO-1 and HO-1 antioxidant enzyme transcription (Figure 3C). Consistent with our hypothesis, the ATO+VEN combination strongly augmented levels of mitochondrial ROS and apoptosis in LSC-enriched progenitor cells, as shown by MitoSOX staining for ROS, Annexin V flow cytometry, DNA fragmentation and cleaved PARP Western blot (Figure 3D). Pre-treatment of the progenitor cells with *N*-acetyl-l-cysteine (NAC), a ROS scavenger, followed by combined treatment with ATO and VEN, reduced ROS levels and inhibited apoptosis, as shown by diminished MitoSOX staining for ROS and attenuated Annexin V flow cytometry staining and DNA fragmentation, both markers of apoptosis (Figure 3E). These results demonstrate that VEN enhances the antileukemic effect of ATO on LSC-enriched progenitor cells at least partly through Nrf2 inhibition. We also performed drug combination studies based on the Chou-Talalay method [48] to quantify the antileukemic effect of ATO and VEN on the LSC-enriched AML progenitor cells. The cells were treated for 72 h with the individual drugs or the ATO/VEN combination, followed by measurement of cell viability. Drug-synergy analysis revealed synergy between ATO and VEN, as shown by the cell viability curves (Appendix A) and isobologram analysis (Figure 3F).

### 2.4. Addition of VEN to ATO Inhibits Mitochondrial Metabolism in LSC-Enriched AML Progenitor Cells

Since interference in mitochondrial metabolism has emerged as a target for AML treatment and could be profoundly altered through mitochondrial ROS induction by ATO+VEN [2,49], we investigated the effects of the ATO+VEN combination on mitochondrial metabolism. Treatment of LSC-enriched progenitor cells with ATO+VEN resulted in significantly decreased fatty acid oxidation (FAO) and oxygen consumption rate (OCR), indicative of OXPHOS disruption (Figure 4A), as compared to single agents. Moreover, ATO+VEN in LSC-enriched progenitor cells resulted in “unhealthy” mitochondria, as shown by an increased ratio of monomeric JC-1 dye (green) versus JC-1 aggregates (J-aggregates, red), representing reduced mitochondrial membrane potential (Figure 4B) and by decreased mitochondria size, as shown by electron microscopy (Figure 4C). Since Nrf2 is involved in regulating mitochondrial ROS, mitochondrial membrane potential, FAO, OXPHOS, and mitochondrial function [50], our results support a molecular mechanism model in which VEN and ATO synergize in LSC-enriched progenitor cells by targeting Nrf2/ROS signaling, leading to impaired mitochondrial metabolism and induction of apoptosis.

### 2.5. Addition of VEN to ATO Inhibits Mitochondrial Metabolism in VEN-Resistant and -Sensitive LSC-Enriched AML Progenitor Cells

Metabolic alterations including a shift from amino-acid driven OXPHOS to FAO-driven OXPHOS have been implicated as mechanisms of VEN resistance in AML [4,11]. Importantly, the ATO+VEN combination also significantly suppressed FAO (Figure 5A), OXPHOS (Figure 5B), and mitochondria size (Figure 5C), and induced apoptosis (Figure 5D) in both VEN-sensitive and VEN-resistant LSC-enriched progenitor cells.

The schematic diagram in Figure 6 represents a proposed antileukemic mechanism of ATO+VEN synergy for inducing ROS and targeting mitochondria metabolism in LSCs. We hypothesize that although ATO induces cytotoxicity through induction of ROS, ATO also simultaneously activates the Nrf2 antioxidant pathway, a major protective mechanism in LSCs to neutralize ROS. Addition of VEN to ATO-treated LSCs can overcome the ROS-attenuating effect of ATO through the inhibitory effect of VEN on Nrf2 signaling. Inhibitors of Akt may further potentiate the cytotoxicity of ATO+VEN. Importantly, the ATO+VEN combination also inhibited mitochondrial metabolism in LSC-enriched progenitor cells from both VEN-sensitive and VEN-resistant AML patients. Thus, our model demonstrates a mechanism for targeting LSCs using two already approved agents for AML, ATO and VEN, and suggests a potential novel treatment option for VEN-resistant AML.

## 3. Materials and Methods

Human samples. Human specimens were collected from patients registered at the City of Hope (COH) National Medical Center who had consented to the City of Hope Institutional Review Board approved protocol (IRB#18067); specimens from healthy donors were collected under COH IRB# 06229. The study was conducted in accordance with the Declaration of Helsinki. Patient characteristics of primary AML samples are listed in Appendix A.

Isolation of mononuclear cells from patient samples. Each patient specimen was transferred to a 50 mL conical tube and the volume was brought up to 25 mL using warm 1x Dulbecco’s phosphate buffered saline (DPBS) with 2% FBS. The specimen was layered on the top of 20 mL Ficoll-Paque Plus in a 50 mL conical tube. Then, the tube was centrifuged at 300× *g* for 32 min continuosly. The layer containing PBMC and plasma was carefully transferred to a 50 mL conical tube and the volume was brought up to 50 mL with warm 1x DPBS. The tube was then centrifuged at 2400 rpm for 8 min. The supernatant was discarded, and the pellet was resuspended in 10 mL of warm DPBS (1x). Cell number and viability were determined, and the sample was frozen. CD34+CD38- cells were then isolated using a magnetic bead selection protocol (Miltenyi Biotech, Bergisch Gladbach, Germany). Patient characteristics are listed in Appendix A.

Chemicals. Venetoclax was purchased from Selleckchem (Houston, TX, USA). Arsenic was purchased from Teva pharmaceuticals (Parsippany, NJ, USA).

DNA fragmentation analysis. Treated cells were lysed on ice for 60 min in 500 μL lysis buffer containing 0.02% SDS, 1% Nonidet P-40 and 0.2 mg/mL proteinase K in PBS. Genomic DNA was extracted using the phenol/chloroform method. The pellet was dissolved in 50 μL of TE buffer (supplemented with 10 mg/mL RNase) for 2 h at 37 °C. A total of 10 μg of DNA was loaded on a 2% agarose gel and visualized under UV light.

FAO assay. Cells were washed with HBSS and incubated with 200 µL of [^3^H]-palmitic acid (1 mCi/mL, Perkin Elmer) bound to fatty-acid free albumin (100 µM; the ratio of palmitate:albumin was 2:1) and 1 mM l-cartinine. The complex was incubated for 2 h at 37 °C. The supernatant was collected after incubation and added to a tube containing 200 µL of cold 10% trichloroacetic acid. The tubes were centrifuged for 10 min at 3000 g at 4 °C and aliquots of supernatants (350 µL) were removed, neutralized with 55 µL of 6 N NaOH and applied to an ion exchange column loaded with Dowex 1 × 2 chloride form resin (Sigma Aldrich, Burlington, MA, USA). The radioactive product was eluted with water. Flow-through was collected and radiation was quantified using liquid scintillation counting.

Seahorse assay. Cells (40,000) in 200 uL cell culture medium were seeded in each well of a XF-96-well cell culture microplate and cultured overnight at 37 °C in 5% CO_2_. As a negative control, three wells were kept devoid of cells and given only Seahorse media, which is comprised of basal XF media, 5.5 mM glucose, 1 mM sodium pyruvate, and 4 mM glutamine (additionally, the pH was adjusted to 7.4). Twelve hours prior to running a plate, the Seahorse sensor cartridge was incubated with Seahorse Calibrant solution according to the manufacturer’s protocol, in a 37 °C, CO_2_-free incubator. On the day of an assay, cells were washed and incubated with Seahorse media. The sensor cartridge was fitted onto the cell culture plate, which was then placed into a 37 °C, CO_2_-free incubator for one hour. During the assay, which was run on the Seahorse XF96 Analyzer, the following inhibitors were injected sequentially, as is standard for the Cell Energy Test: oligomycin (1 mM), FCCP (0.5 mM).

Immunoprecipitation and immunoblotting analyses. Cells were washed in ice-cold PBS and lysed in buffer containing 1 mM phenylmethanesulfonyl fluoride and 10 mM protease inhibitor cocktail. For immunoprecipitation, 500 μg of cell lysate was incubated with indicated antibodies overnight at 4 °C. 30 μL of Protein A/G agarose beads (Calbiochem) were added, and the mixture was inverted for 2 h at 4 °C. For immunoblotting, the immunoprecipitated complex or 30 μg of each cell lysate was separated on NuPAGE 4–12% gradient gels (Invitrogen, Waltham, MA, USA) and immunocomplexes were visualized with enhanced chemiluminescence reagent (Thermo Scientific, Lafayette, CO, USA). Antibodies used for IP and IB analysis are listed in Appendix A.

Measurement of ROS. For quantification of ROS, cells were incubated with 3 μM of MitoSOX Red (Life Technologies—Molecular Probes, Eugene, OR, USA) in culture medium for 30 min at 37 °C. Cells were then washed with PPBS and stained for Annexin V/DAPI analysis by flow cytometry. Only the live cell population (Annexin-V-/DAPI-) was analyzed for ROS production. To determine the effect of NAC (*N*-acetyl-L-cysteine), an ROS scavenger, cells were cultured with and without 2.5 mM of NAC for 2 h, followed by ATO and VEN combined treatment for additional 12 h.

Cellular Fractionation. Cells were collected and washed in PBS followed by fractionation into nuclear and cytoplasmic fractions using a subcellular fractionation kit (Thermo Scientific, Lafayette, CO, USA). Briefly, the cells were vigorously vortexed in cytoplasmic extraction reagents and centrifuged to isolate the soluble cytoplasmic fraction. The remaining insoluble fraction, which contained nuclei, was suspended in nuclear extraction reagent and centrifuged to collect the nuclear fraction. All steps were performed at 4 °C.

RT-PCR and q-PCR analysis. To measure mRNA expression, total RNA was extracted using the RNeasy Mini Kit (Qiagen, Valencia, CA, USA). First-strand cDNA was synthesized using the SuperScript III First-Strand Kit and q-PCR was performed using TaqMan Gene Expression Assays (Thermo Fisher). Results are presented as log2-transformed ratio according to the 2^−ΔCt^ method (ΔCt = Ct of target −Ct of reference). Primer sequences used for qPCR analysis are shown in Appendix A.

Synthetic small interfering RNA (siRNA) oligonucleotides. The siGENOME SMARTpool for siRNA of AKT and Nrf2 was purchased from Thermo Scientific (Lafayette, CO, USA). Scrambled control RNA (siSCR) was used as a control. The target sequences for siRNAs are shown in Appendix A.

Assessing apoptosis using flow cytometry. The Annexin-V and DAPI double staining method was used to detect apoptosis. Cells were harvested and washed twice with Annexin-V binding buffer (BD Bioscience, San Jose, CA, USA) and resuspended in 100 μL of the same buffer containing Annexin-V APC (BD Bioscience, San Jose, CA, USA). Cells were then incubated in the dark at room temperature for 15 min, washed again and resuspended in 300 μL of buffer. DAPI (Sigma-Aldrich) was added immediately before analysis by LSR II flow cytometer (BD Bioscience, San Jose, CA, USA).

Immunocytochemistry. Cells were collected, washed in ice-cold PBS and mounted on glass slides using a Cytocentrifuge (CytoSpin4, 600 rpm, 10 min). Cells were then washed with PBS, fixed in 4% paraformaldehyde for 15 min and permeabilized in 0.5% Triton X-100 for 15 min. Non-specific epitopes were blocked with 5% bovine serum albumin (BSA) for 30 min. Primary antibodies are listed in Appendix A. Secondary anti-mouse/rabbit/goat-Alexa 594/488/647 goat antibodies were purchased from Thermo Scientific (Lafayette, CO, USA). Cell images were acquired using a Zeiss confocal laser-scanning-microscope (Zeiss LSM 800). Nuclei were counterstained with ProLong Gold Antiface with DAPI (Molecular Probes, Invitrogen, Carlsbad, CA, USA).

Transmission electron microscopy. Cultured cells were fixed with 2.5% glutaraldehyte, 0.1 M Cacodylate buffer (Na(CH_3_)_2_AsO_2_ 3H_2_O), pH 7.2, at 4 °C. Standard sample preparation for TEM was followed, including post-fixation with osmium tetroxide, serial dehydration with ethanol, and embedment in Eponate. Ultra-thin sections (70 nm thick) were acquired by ultramicrotomy, post-stained, and examined on an FEI Tecnai 12 transmission electron microscope equipped with a Gatan OneView CMOS camera. TEM images were taken at nominal 11,000× magnification.

Measurement of mitochondrial membrane potential. Mitochondrial membrane potential in cells was visualized by staining with JC-1 dye (Cat# T3168, ThermoFisher, Waltham, MA USA) using a confocal microscope (LSM880, Zeiss, Jena, Germany). Briefly, cells were collected, washed in ice-cold PBS and mounted on glass slides using a Cytocentrifuge (CytoSpin4, 600 rpm, 10 min). Cells were then washed with PBS, fixed in 4% paraformaldehyde for 15 min and permeabilized in 0.5% Triton X-100 for 15 min. Cells were then stained with JC-1 dye for 1 h at 37 °C. JC-1 dye exhibits potential-dependent accumulation in mitochondria, indicated by green fluorescence emission at (~529 nm) for the monomeric form of JC-1, which shifts to red (~590 nm) with a concentration-dependent formation of red fluorescent JC-1 aggregates (J-aggregates). Consequently, mitochondrial depolarization is indicated by a decrease in the red/green fluorescence intensity ratio (ThermoFisher).

Combined drug effect analysis. For two-drug combination experiments, the cells were treated with VEN or ATO for 72 h, as single agents as well as in combination, at constant ratios, on the basis of the previously calculated IC_50_ values for each drug, as indicated. Quantitative analysis of dose-effect relationships was determined after measurement of cell growth using MTS assay. Potential synergistic or additive effects were calculated using the software CompuSyn, Cambridge, UK). Isobolograms and combination-index plots (not shown) were created, and combination index (CI) values calculated. Drug synergism, addition, and antagonism effects are defined by CI values of <1.0, 1.0, and >1.0, respectively.

Statistical analysis: To compare the means of two groups, results were compared using unpaired, two-tailed Student’s t test, with values from at least two independent experiments, in triplicates. Data are presented as mean  ±  standard error, as indicated. *p* < 0.05 was considered statistically significant.

## 4. Conclusions

The present study elucidates the synergistic antileukemic mechanism of the ATO+VEN combination in targeting LSC-enriched AML progenitor cells. ATO induces oxidative stress by increasing ROS levels in cells, and high ROS levels activate the Nrf2 antioxidant pathway to counteract ROS. It is still controversial whether activation or inhibition of the Nrf2 antioxidant pathway is beneficial for cancer therapy [51] and likely also depends on the cancer cell type. We show that the ATO+VEN combination induces augmented ROS signaling through down-regulation of the Akt-Nrf2 antioxidant pathway by VEN, which was associated with decreased FAO and OXPHOS and increased apoptosis and DNA fragmentation in LSC-enriched AML progenitor cells. Importantly, the ATO+VEN combination significantly inhibited Nrf2 signaling and promoted apoptosis in LSC-enriched blast cells derived from both VEN-sensitive and VEN-resistant AML patients. Based on our results and the results from a previous in vitro study with ATO+VEN in AML cell lines and primary cells [15] we provide additional rationale for advancing this combination to preclinical and clinical trials in AML, including in VEN-resistant AML. In summary, the ATO+VEN combination is a promising novel regimen for treatment of AML.

## Figures and Tables

**Figure 1 ijms-23-06568-f001:**
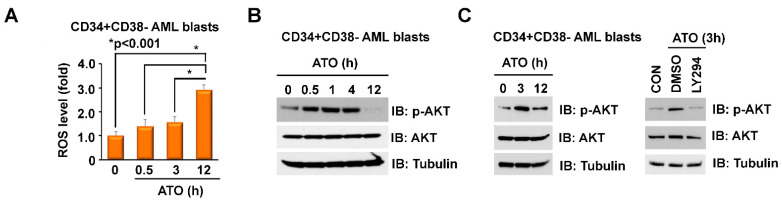
ATO induces ROS and p-AKT in LSC-enriched AML progenitor cells. (**A**–**C**) Effects of ATO on levels of Akt phosphorylation and mitochondrial ROS in LSC-enriched progenitor cells. LSC-enriched progenitor cells (CD34+CD38- enriched blast cell population, *n* = 5, pooled samples) were treated with 500 nM ATO for up to 12 h, as indicated. (**A**) ROS levels were measured by flow cytometry using mitoSox staining. (**B**) Cell lysate was immunoblotted with indicated antibodies. (**C**) Left, LSC-enriched progenitor cells were treated with for 3 and 12 h. the indicated times. Right, LSC-enriched progenitor cells were co-treated with 500 nM ATO and DMSO or 10 µM LY294002 for 3 h. Cell lysate was immunoblotted with indicated antibodies.

**Figure 2 ijms-23-06568-f002:**
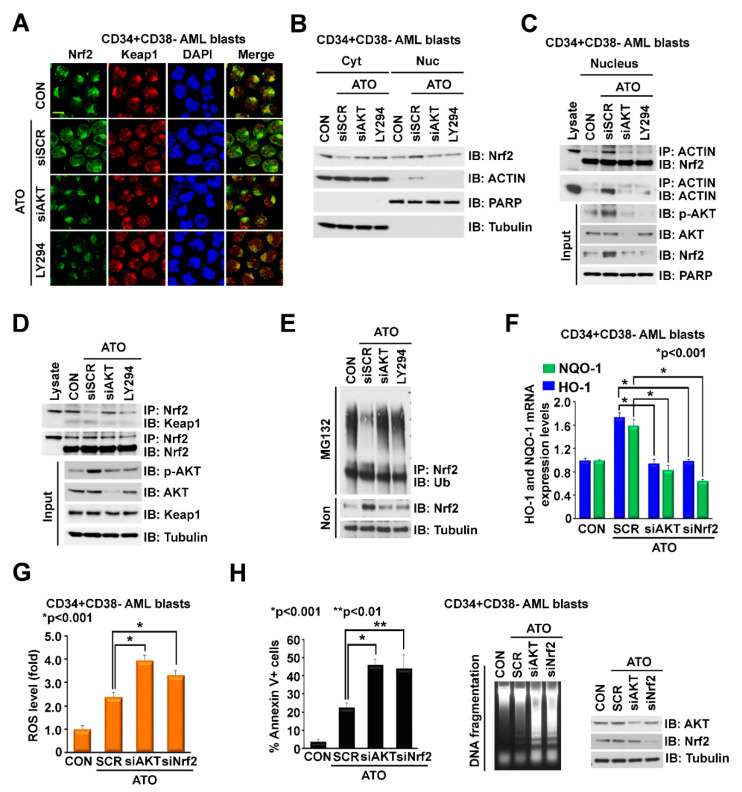
ATO-induced ROS activation is attenuated by concomitant activation of the p-Akt/Nrf2 antioxidant pathway in LSC-enriched AML progenitor cells. (**A**,**B**) Effects of ATO on cellular distribution of Nrf2 and Actin. LSC-enriched progenitor cells were treated with 500 nM ATO in the presence of siSCR control, siAkt (20 nM), or LY294002 (10 µM) for 12 h. (**A**) Cells were stained with anti-Nrf2 and anti-Keap1 antibodies. (**B**) Cells were fractionated into cytoplasmic and nuclear parts and lysates were immunoblotted with the indicated antibodies. (**C**) Effects of ATO on Nrf2/Actin interaction in the nucleus. LSC-enriched progenitor cells were treated and fractionated as described in (**A**,**B**). Nuclear lysate was immunoprecipitated with anti-Actin and immunoblotted with anti-Nrf2 antibodies. Input controls are shown. (**D**,**E**) Effects of ATO on Nrf2-Keap1 binding and Keap1-regulated Nrf2 ubiquitylation. LSC-enriched progenitor cells were treated as described in (**A**,**B**). (**D**) Cell lysate was immunoprecipitated with anti-Nrf2 and immunoblotted with anti-Keap1 antibodies. Input controls are shown. (**E**) Cell lysate was immunoprecipitated with anti-Nrf2 and immunoblotted with anti-Ubiquitin (Ub) antibodies. (**F**–**H**) Effects of Akt or Nrf2 KD on ATO-regulated levels of NQO-1/HO-1 expression, ROS and apoptosis in LSC-enriched progenitor cells. LSC-enriched progenitor cells were treated with 500 nM ATO in the presence of siSCR, siAkt (20 nM) or siNrf2 (20 nM), for 24 h. (**F**) mRNA levels of NQO-1 and HO-1 (normalized to GAPDH). (**G**) ROS levels measured by mitoSox staining. (**H**) Apoptosis levels determined by Annexin V flow cytometry (left) and genomic DNA fragmentation (middle). Akt and Nrf2 KD efficacy is shown on the right. Scale bar, 10 µm.

**Figure 3 ijms-23-06568-f003:**
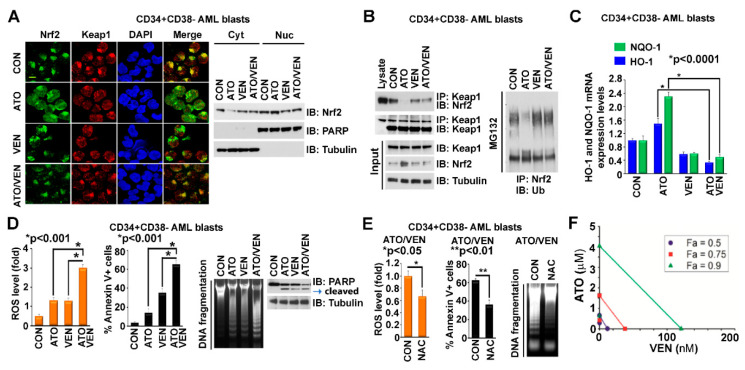
Addition of VEN to ATO inhibits Nrf2 nuclear translocation, augments ROS and increases apoptosis in LSC-enriched AML progenitor cells. (**A**) Effects of VEN on ATO-induced Nrf2 translocation. LSC-enriched progenitor cells were treated with 500 nM ATO, 10 nM VEN, or both. Left, the cells were stained with anti-Nrf2 and anti-Keap1 antibodies. Scale bar, 10 μm. Right, the cells were fractionated into cytoplasmic and nuclear parts and immunoblotted with indicated antibodies. (**B**) Effects of VEN on ATO-regulated Nrf2/Keap1 binding and Nrf2 ubiquitylation. LSC-enriched progenitor cells were treated as described in (**A**). Left, cell lysate was immunoprecipitated with anti-Keap1 and immunoblotted with anti-Nrf2 antibodies. Input controls are shown. Right, cell lysate was immunoprecipitated with anti-Nrf2 and immunoblotted with anti-Ub antibodies. (**C**,**D**) Effects of VEN on ATO-regulated levels of NQO-1/HO-1 expression, mitochondrial ROS and apoptosis in LSC-enriched progenitor cells treated as described in (**A**). (**C**) mRNA levels of NQO-1 and HO-1 (normalized to GAPDH). (**D**) Left, mitochondrial ROS levels. Middle and right, apoptosis levels by Annexin V flow cytometry, genomic DNA fragmentation, and PARP cleavage. (**E**) Effects of NAC pre-treatment on ATO+VEN-regulated levels of ROS and apoptosis in LSC-enriched progenitor cells. Cells were treated with ATO (500 nM) and VEN (10 nM) in the presence of DMSO control or NAC (2.5 mM). Left, mitochondrial ROS levels. Middle and right, apoptosis levels by Annexin V flow cytometry and DNA fragmentation. (**F**) Synergistic effects of ATO and VEN on cell viability. LSC-enriched progenitor cells were cultured with ATO, VEN or ATO+VEN, at increasing concentrations, for 72 h, followed by MTS assay. The synergistic interaction between VEN and ATO was analyzed using CalcuSyn program and isobolographic representation of the combination is shown. The individual doses of VEN and ATO to achieve 90% (green line), 75% (red line), and 50% (dark blue line) growth inhibition were plotted on the *x* and *y* axes. Combination index values are represented by points above (indicating antagonism) or below the lines (indicating drug synergy). The isobologram demonstrates drug synergy for all dose combinations shown.

**Figure 4 ijms-23-06568-f004:**
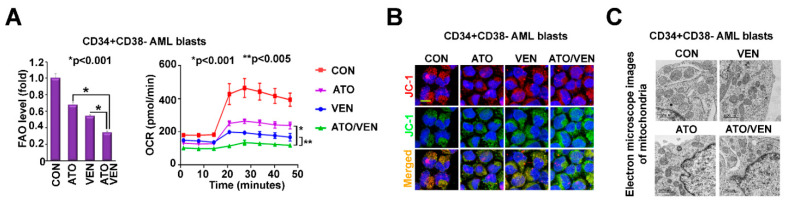
Addition of VEN to ATO inhibits mitochondrial metabolism in LSC-enriched AML progenitor cells. (**A**) Effects of the ATO and VEN combination on levels of FAO and OXPHOS. LSC-enriched progenitor cells were treated with 500 nM ATO, 10 nM VEN, or both. Left, levels of FAO were measured by the oxidation rate of 3H-palmitic acid. Right, levels of oxidative consumption rate (OCR) were measured by Seahorse cell energy testing assay. (**B**,**C**) Effects of the ATO and VEN combination on mitochondrial membrane potential and mitochondria morphology in LSC-enriched progenitor cells. The cells were treated as described in (**A**). (**B**) Mitochondrial membrane potential was measured by staining of the treated cells with JC-1 dye. The red and green fluorescence indicates JC-1 polymers and monomers, respectively. Scale bar, 10 µm. (**C**) Mitochondria morphology was measured by electron microscope imaging. Scale bar, 1000 nm.

**Figure 5 ijms-23-06568-f005:**
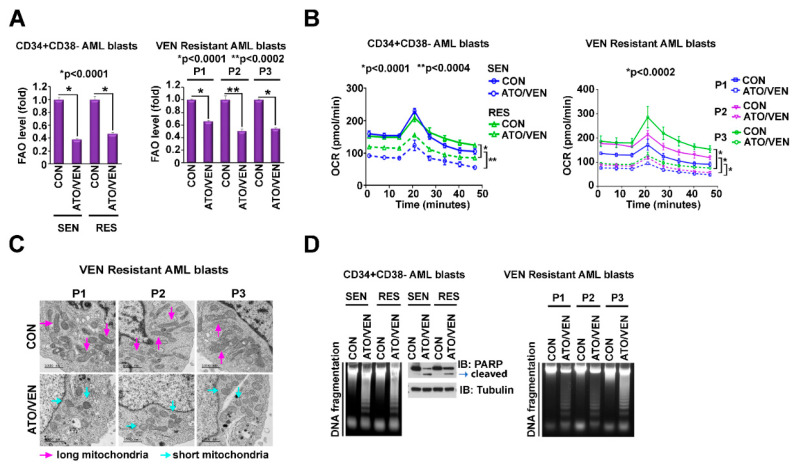
Addition of VEN to ATO inhibits mitochondrial metabolism in VEN-resistant and -sensitive LSC-enriched AML progenitor cells. (**A**–**D**) Effects of ATO plus VEN on OXPHOS metabolism and apoptosis in VEN-sensitive and -resistant LSCs. CD34+CD38-cells isolated from VEN-sensitive (*n* = 3) and VEN-resistant (*n* = 3) primary AML patient samples were treated for 24 h with 10 nM VEN plus 500 nM ATO. (**A**) Levels of FAO. The levels of FAO of pooled samples (each, *n* = 3) are shown on the left and FAO levels of individual samples of VEN-resistant AML cells are shown on the right. (**B**) Levels of OCR. The levels of OCR of pooled samples (each, *n* = 3) are shown on the left and OCR levels of individual samples of VEN-resistant AML cells are shown on the right. (**C**) Mitochondria morphology of DMSO-treated or ATO+VEN-treated VEN-resistant primary AML cells was measured by electron microscope imaging. Red arrow, long (healthy) mitochondria. Green arrow, short (unhealthy) mitochondria. Scale bar, 1000 nm. (**D**) Levels of apoptosis. The levels of apoptosis demonstrated by DNA fragmentation and PARP cleavage of pooled samples are shown on the left and apoptosis levels of individual samples of VEN-resistant AML cells are shown on the right. FAO, fatty acid oxidation; OXPHOS, oxidative phosphorylation; VEN, venetoclax; ATO, arsenic trioxide.

**Figure 6 ijms-23-06568-f006:**
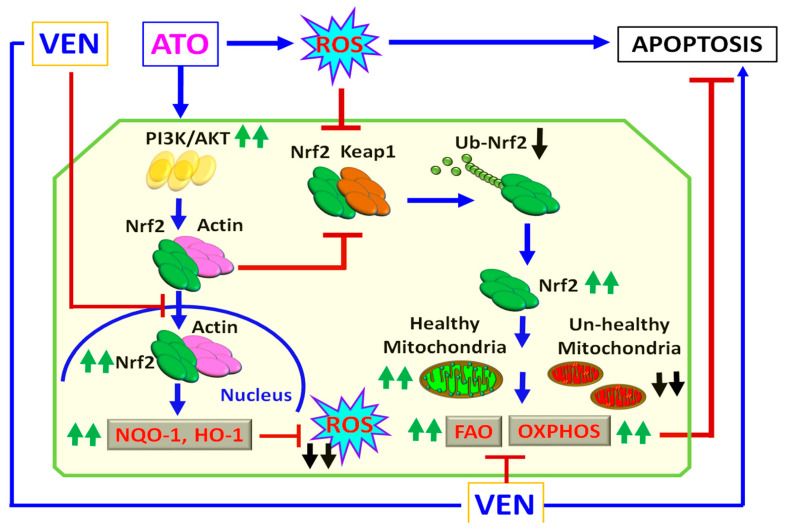
Antileukemic mechanism of ATO +VEN synergy. Schematic diagram of a proposed mechanism of synergy of combining ATO with VEN for targeting of metabolic vulnerabilities of LSCs. FAO, fatty acid oxidation; OXPHOS, oxidative phosphorylation; VEN, venetoclax; ATO, arsenic trioxide. Blue arrows, activation; red lines, inhibition; green arrows, increase; black arrows, decrease.

## Data Availability

Raw data/analyzed raw data used in the current study are available from the corresponding authors on reasonable request.

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
