# Peer review of "Arsenic Trioxide and Venetoclax Synergize against AML Progenitors by ROS Induction and Inhibition of Nrf2 Activation"

_ijms, 2022, doi:10.3390/ijms23126568_

Round 1
Reviewer 1 Report
The search for and modification of chemotherapeutic drugs is an urgent task of modern pharmacology, since resistance is formed with prolonged use of antitumor agents. The use of combinations of several drugs can act as a deterrent to the development of resistance.
The article is generally well written, but there are a few caveats:
-Drawings 1 and 2 are very cluttered, it is advisable to divide each drawing into 2-3 and make your own signature for each. In the presented format, the information is poorly perceived, the drawings, especially photographs, are unreadable, as well as the description of the results in one place, the drawings in another, the caption to the drawings is huge.
-For Figure 3, expand the caption a little, explain the meaning of the dotted line, arrows (of different colors), and so on.
-Figure 2J, 2G and 2K have different error values (*). Why are there "*" everywhere on different values?
-In Figures 1A, 1J, 1K, 1I and 2C, 2D, 2E, the “*” values are not indicated.
Reviewer 2 Report
Generally the study presented in the manuscript looks interested and valuable to the readers of the journal. However following minor corrections are suggested for the improvement of the article.
1. The Abstract should be reviewed again with answers to the following questions: What problem was studied and why is it important? What methods were used? What are the important results? What conclusions can be drawn from the results? What is the novelty of the work and where does it go beyond previous efforts in the literature?
2. Introduction is very weeks and presented with discussion on only 10 articles. The relevant literature review is missing, Problem statement and justification need further elaboration. The innovative contributions, objective and insight should be the part of introduction (may list at the end of the introduction).
3. Material and methods section is missing and need to the part of the manuscript.
4. Further elaborative description of each graphical and numerical illustration should be given in the results section.
5. Advantages/limitation and future scope of the work should be given in the conclusion section.
Reviewer 3 Report
This is very interesting paper concluding with therapeutic approach for acute myeloid leukemia (AML). Authors have measured several parameters to justify the findings. This study shows very interesting approach for the treatment in combination of arsenic trioxide (ATO) with venetoclax (VEN). I have some minor comments-
- In the introduction section, several recent studies on arsenic trioxide are missing. Authors should update.
- Overall the presentation and description of the results are quite clear. I would like to know that in Figure 1a, i, j, k, did they compare with control group? If yes, then why significance levels are not indicated in comparation with control as this is quite clear in figure 2(J).
- In figure 2L, indicate changes with an appropriate arrows.
- The methods section should be included in the main manuscript file. Atleast, it should be defined in short. It would be nice to add a flow diagram indicating the overall methodology obtained in the study, in case if not giving full description.
- Author should add conclusion in the manuscript after results and discussion section.
Round 2
Reviewer 2 Report
Revised manuscript improve and recommended